# Q-Ensemble for Offline RL: Don't Scale the Ensemble, Scale the Batch Size

**Alexander Nikulin**
Tinkoff
a.p.nikulin@tinkoff.ai

**Vladislav Kurenkov**
Tinkoff
v.kurenkov@tinkoff.ai

**Denis Tarasov**
Tinkoff
den.tarasov@tinkoff.ai

**Dmitry Akimov**
Tinkoff
d.akimov@tinkoff.ai

**Sergey Kolesnikov**
Tinkoff
s.s.kolesnikov@tinkoff.ai

## Abstract

Training large neural networks is known to be time-consuming, with the learning duration taking days or even weeks. To address this problem, large-batch optimization was introduced. This approach demonstrated that scaling mini-batch sizes with appropriate learning rate adjustments can speed up the training process by orders of magnitude. While long training time was not typically a major issue for model-free deep offline RL algorithms, recently introduced Q-ensemble methods achieving state-of-the-art performance made this issue more relevant, notably extending the training duration. In this work, we demonstrate how this class of methods can benefit from large-batch optimization, which is commonly overlooked by the deep offline RL community. We show that scaling the mini-batch size and naively adjusting the learning rate allows for (1) a reduced size of the Q-ensemble, (2) stronger penalization of out-of-distribution actions, and (3) improved convergence time, effectively shortening training duration by 3-4x times on average.

## 1 Introduction

Offline Reinforcement Learning (ORL) provides a data-driven perspective on learning decision-making policies by using previously collected data without any additional online interaction during the training process (Lange et al., 2012; Levine et al., 2020). Despite its recent development (Fujimoto et al., 2019; Nair et al., 2020; An et al., 2021; Zhou et al., 2021; Kumar et al., 2020) and application progress (Zhan et al., 2022; Apostolopoulos et al., 2021; Soares et al., 2021), one of the current challenges in ORL remains algorithms extrapolation error, which is an inability to correctly estimate the values of unseen actions (Fujimoto et al., 2019). Numerous algorithms were designed to address this issue. For example, Kostrikov et al. (2021) (IQL) avoids estimation for out-of-sample actions entirely. Similarly, Kumar et al. (2020) (CQL) penalizes out-of-distribution actions such that their values are lower-bounded. Other methods explicitly make the learned policy closer to the behavioral one (Fujimoto & Gu, 2021; Nair et al., 2020; Wang et al., 2020).

In contrast to prior studies, recent works (An et al., 2021) demonstrated that simply increasing the number of value estimates in the Soft Actor-Critic (SAC) (Haarnoja et al., 2018) algorithm is enough to advance state-of-the-art performance consistently across various datasets in the D4RL benchmark (Fu et al., 2020). Furthermore, An et al. (2021) showed that the double-clip trick actually serves as an uncertainty-quantification mechanism providing the lower bound of the estimate, and simply increasing the number of critics can result in a sufficient penalization for out-of-distribution actions. Despite its state-of-the-art results, the performance gain for some datasets requires significant

3rd Offline Reinforcement Learning Workshop at Neural Information Processing Systems, 2022.

computation time or optimization of an additional term, leading to extended training duration (Figure 2).

In this paper, inspired by parallel works on reducing the training time of large models in other areas of deep learning (You et al., 2019, 2017) (commonly referred to as large batch optimization), we study the overlooked use of large batches[1] in the ORL setting. We demonstrate that, instead of increasing the number of critics or introducing an additional optimization term in SAC-N (An et al., 2021) algorithm, simple batch scaling and naive adjustment of the learning rate can (1) provide a sufficient penalty on out-of-distribution actions and (2) match state-of-the-art performance on the D4RL benchmark. Moreover, this large batch optimization approach significantly reduces the convergence time, making it possible to train models 4x faster on a single-GPU setup. To the best of our knowledge, this is the first study that examines large batch optimization in the ORL setup.

## 2 Q-Ensemble For Offline RL

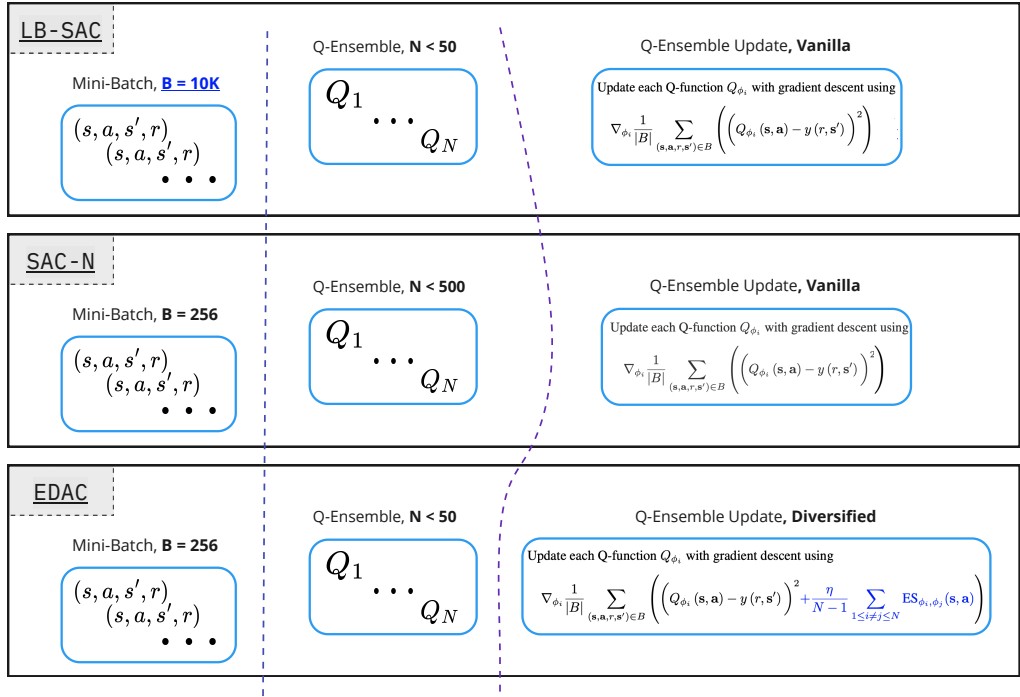

Figure 1: The difference between recently introduced SAC-N, EDAC, and the proposed LB-SAC. The introduced approach does not require an auxiliary optimization term while making it possible to effectively reduce the number of critics in the Q-ensemble by switching to the large-batch optimization setting.

Ensembles have a long history of applications in the reinforcement learning community. They are employed in the model-based approaches to combat compounding error and model exploitation (Kurutach et al., 2018; Chua et al., 2018; Lai et al., 2020; Janner et al., 2019), in model-free to greatly increase sample efficiency (Chen et al., 2021; Hiraoka et al., 2021; Liang et al., 2022) and in general to boost exploration in online RL (Osband et al., 2016; Chen et al., 2017; Lee et al., 2021; Ciosek et al., 2019). In offline RL, ensembles were mostly utilized to model epistemic uncertainty in value function estimation (Agarwal et al., 2020; Bai et al., 2022; Ghasemipour et al., 2022), introducing uncertainty aware conservatism.

Recently, An et al. (2021) investigated an isolated effect of clipped Q-learning on value overestimation in offline RL, increasing the number of critics in the Soft Actor Critic (Haarnoja et al., 2018) algorithm from 2 to $N$. Surprisingly, with tuned $N$ SAC-N outperformed previous state-of-the-art algorithms

---

[1]Offline RL is often referred to as Batch RL, here, we extensively use the *batch* term to denote a mini-batch, not the dataset size.

on D4RL benchmark (Fu et al., 2020) by a large margin, although requiring up to 500 critics on some datasets. To reduce the ensemble size, An et al. (2021) proposed EDAC which adds auxiliary loss to diversify the ensemble, allowing to greatly reduce $N$ (Figure 1).

Such pessimistic Q-ensemble can be interpreted as utilizing the Lower Confidence Bound (LCB) of the Q-value predictions. Assuming that $Q(s, a)$ follows a Gaussian distribution with mean $m$, standard deviation $\sigma$ and $\{Q_j(s, a)\}_{j=1}^{N}$ are realizations of $Q(s, a)$, we can approximate expected minimum (An et al., 2021; Royston, 1982) as

$$\mathbb{E}\left[\min_{j=1,\dots,N} Q_j(s, a)\right] \approx m(s, a) - \Phi^{-1}\left(\frac{N - \frac{\pi}{8}}{N - \frac{\pi}{4} + 1}\right)\sigma(s, a) \tag{1}$$

where $\Phi$ is the CDF of the standard Gaussian distribution. In practice, OOD actions have higher variance on Q-value estimates compared to ID (Figure 5a). Thus, with increased $N$ we strengthen value penalization for OOD actions, inducing conservatism. As we will show in Section 5.2, this effect can be amplified by scaling batch instead of ensemble size.

## 3 How Long Does It Take for Q-Ensemble to Converge?

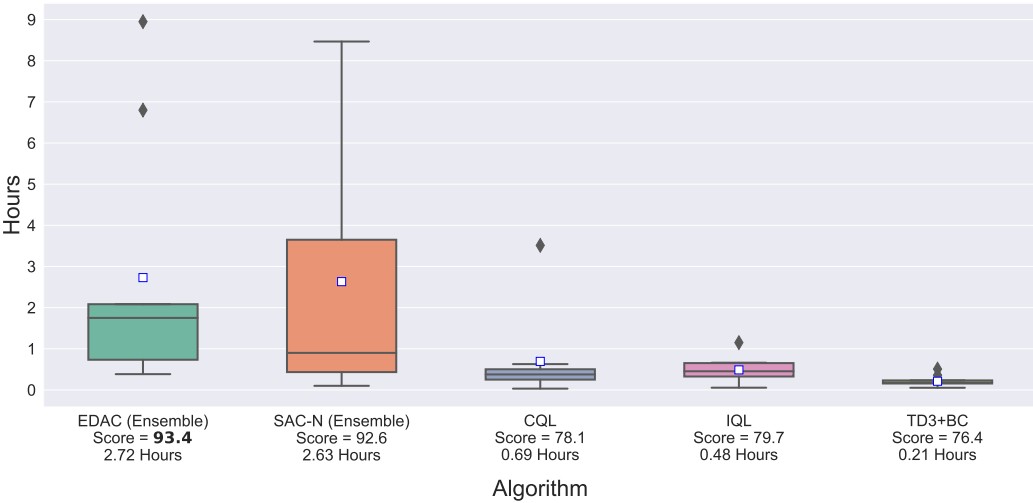

Figure 2: The convergence time of popular deep offline RL algorithms on 9 different D4RL locomotion datasets (Fu et al., 2020). We consider algorithm convergence similar to Reid et al. (2022), i.e., marking convergence at the point of achieving results similar to the ones reported in performance tables. White boxes denote mean values (which are also illustrated on the x-axis). Black diamonds denote samples. Note that the convergence time of Q-ensemble based methods is significantly higher. Raw values can be found in the Appendix A.6.

To demonstrate the inflated training duration of Q-ensemble methods, we start with an analysis of their convergence time compared to other deep offline RL algorithms. Here, we show that while these methods consistently achieve state-of-the art performance across many datasets, the time it takes to obtain such results is significantly longer when compared to their closest competitors.

While it is not a common practice to report training times or convergence speed, some authors carefully analyzed these characteristics of their newly proposed algorithms (Fujimoto & Gu, 2021; Reid et al., 2022). There are two prevailing approaches for this analysis. First is to study the *total training time* for a fixed number of training steps or the speed per training step (Fujimoto & Gu, 2021; An et al., 2021). Second approach, such as the one taken by Reid et al. (2022), is to measure the *convergence time* using a relative wall-clock time to achieve the results reported in the performance tables.

Measuring the total training time for a fixed number of steps can be considered a straightforward approach. However, it does not take into account that some algorithms may actually require a smaller (or bigger) number of learning steps. Therefore, in this paper, we choose the second approach, which involves measuring the relative clock-time until convergence. Similar to Reid et al. (2022), we deem

an algorithm to converge when its evaluation (across multiple seeds) achieves a normalized score within two points[2] or higher of the one reported for the corresponding method in the performance table.

We carefully reimplemented all the baselines in same codebase and re-run the experiments to make sure that they are executed on the same hardware (Tesla A100) in order for the relative-clock time to be comparable. The results are depicted in Figure 2. One can see that IQL and TD3+BC are highly efficient compared to their competitors, and their convergence times do not exceed two hours even in the worst case scenarios. This efficiency comes from the fact that these methods do not take long for one training step (Fujimoto & Gu, 2021; Kostrikov et al., 2021). However, while being fast to train, they severely underperform when compared to the Q-ensemble based methods (SAC-N, EDAC).

Unfortunately, this improvement comes at a cost. For example, SAC-N median convergence time is under two hours (2-3x times longer than for IQL or TD3+BC), but some datasets may require up to nine hours of training. This is due to its usage of a high number of critics, which requires more time for both forward and backward passes. Notably, EDAC was specifically designed to avoid using a large number of critics, reporting smaller computational costs per training step (An et al., 2021) and lower memory consumption. However, as can be seen in Figure 2, its convergence time is still on par with the SAC-N algorithm. As we discussed earlier, some algorithms might require a higher number of training iterations to converge, which is exactly the case for the EDAC algorithm.

## 4    Large Batch Soft Actor-Critic

One common approach for reducing training time of deep learning models is to use large-batch optimization (Hoffer et al., 2017; You et al., 2017, 2019). In this section, we investigate a similar line of reasoning for deep ORL and propose several adjustments to the SAC-N algorithm (An et al., 2021) in order to leverage large mini-batch sizes:

**1. Scale mini-batch** Instead of using a significantly larger number of critics, as is done in SAC-N, we instead greatly increase the batch size from the typically used 256 state-action-reward tuples to 10k. Note that in the case of commonly employed D4RL datasets and actor-critic architectures, this increase does not require using more GPU devices and can be executed on the single-GPU setup. While higher batch sizes are also viable, we will demonstrate in Section 5 that this value is sufficient, and does not result in over-conservativeness.

**2. Learning rate square root scaling** In other research areas, it was often observed that a simple batch size increase can be harmful to the performance, and other additions are needed (Hoffer et al., 2017). The learning rate adjustment is one such modification. Here, we use the Adam optimizer (Kingma & Ba, 2014), fixing the learning rate derived with a formula similar to Krizhevsky (2014) also known as "square root scaling":

$$learning\ rate = base\ learning\ rate * \sqrt{\frac{Batch\ Size}{Base\ Batch\ Size}} \tag{2}$$

where we take both $base\ learning\ rate$ and $base\ batch\ size$ to be equal to the values used in the SAC-N algorithm. Note that they are the same across all datasets. The specific values can be found in the Appendix A.5. Unlike Hoffer et al. (2017), we do not use a warm-up stage.

We refer to the described modifications as Large-Batch SAC (LB-SAC)[3]. Overall, the resulting approach is equivalent to the SAC-N algorithm, but with carefully adjusted values which are typically considered hyperparameters. Figure 1 summarizes the distinctive characteristics of LB-SAC and the recently proposed deep offline Q-ensemble based algorithms.

---

[2]We derive this value as a mean standard deviation typically observed when evaluating checkpoints across random seeds, e.g., see Table 1.

[3]One may rightfully point out that, while we work in a single-GPU setup, we rely on modern computational devices such as A100 with large memory capabilities. However, as we will demonstrate in the next section, the memory consumption on D4RL datasets does not exceed 5GB of video memory in the worst-case scenarios, i.e., making it possible to utilize devices with less computation power.

# 5 Experiments

In this section, we present the empirical results of comparing LB-SAC with other Q-ensemble methods. We demonstrate that LB-SAC significantly improves convergence time while matching the final performance of other methods, and then analyze what contributes to such performance.

## 5.1 Evaluation on the D4RL MuJoCo Gym Tasks

We run our experiments on the commonly used MuJoCo locomotion subset of the D4RL benchmark (Fu et al., 2020): Hopper, HalfCheetah, and Walker2D. Similar to the experiments conducted in Section 3, we use the same GPU device and codebase for all the comparisons.

**LB-SAC Normalized Scores** First, we report the final scores of the introduced approach. The results are illustrated in Table 1. We see that the resulting scores outperform the original SAC-N algorithm and match the EDAC scores. Moreover, when comparing on the whole suit of locomotion datasets, the average final performance is above both SAC-N and EDAC results (see Table 3 in the Appendix). We highlight this result, as it is a common observation in large-batch optimization that naive learning rate adjustments typically lead to learning process degradation, making adaptive approaches or a warm-up stage necessary (Hoffer et al., 2017; You et al., 2017, 2019). Evidently, this kind of treatment can be bypassed in the ORL setting for the SAC-N algorithm.

Table 1: Final normalized scores on D4RL Gym tasks, averaged over 4 random seeds. LB-SAC outperforms SAC-N and attains a similar performance to the EDAC algorithm while being considerably faster to converge as depicted in Figure 3. For ensembles, we provide additional results on more datasets in the Appendix A.4.

| Task Name | TD3+BC | IQL | CQL | SAC-N | EDAC | LB-SAC |
|---|---|---|---|---|---|---|
| halfcheetah-medium | 48.1 | 48.3 | 47.0 | 67.5±1.2 | 65.9±0.6 | 71.5±1.2 |
| halfcheetah-medium-expert | 90.7 | 94.5 | 95.9 | 107.1±2.0 | 106.3±0.6 | 109.1±2.6 |
| halfcheetah-medium-replay | 44.8 | 43.5 | 45.1 | 63.9±0.8 | 61.3±1.9 | 64.3±0.7 |
| hopper-medium | 60.3 | 62.7 | 64.9 | 100.3±0.3 | 101.6±0.6 | 100.7±5.2 |
| hopper-medium-expert | 101.1 | 106.2 | 93.8 | 110.1±0.3 | 110.7±0.1 | 110.9±0.4 |
| hopper-medium-replay | 64.4 | 84.5 | 87.6 | 101.8±0.5 | 101.0±0.5 | 101.6±1.0 |
| walker2d-medium | 82.7 | 84.0 | 80.3 | 87.9±0.2 | 92.5±0.8 | 90.1±1.4 |
| walker2d-medium-expert | 110.0 | 111.6 | 109.6 | 116.7±0.4 | 114.7±0.9 | 113.4±3.4 |
| walker2d-medium-replay | 85.6 | 82.5 | 79.2 | 78.7±0.7 | 87.1±2.3 | 79.9±0.4 |
| Average | 76.4 | 79.7 | 78.1 | 92.6 | **93.4** | **93.4** |

**LB-SAC Convergence Time** Second, we study the convergence time in a similar fashion to Section 3. The results are depicted in Figure 3. One can see that the training times of LB-SAC are less than those of both EDAC and SAC-N, outperforming them in terms of average and the worst convergence times. Moreover, LB-SAC even outperforms CQL in the worst-case scenario, achieving commensurate average convergence time. This improvement comes from both (1) the usage of low number of critics, similar to EDAC (see Figure 4) and (2) improved ensemble convergence rates, which we demonstrate further in Section 5.2. Additionally, we report convergence times for different return thresholds on all D4RL Gym datasets in the Appendix A.4 (Figure 13) confirming faster convergence to each of them.

**LB-SAC Memory Consumption** Memory consumption is an obvious caveat of using large batch sizes. Here, we report the worst-case memory requirements for all of the Q-ensemble algorithms. Table 2 shows that utilization of large batches can indeed lead to a considerable increase in memory usage. However, this still makes it possible to employ LB-SAC even using single-GPU setups on devices with moderate computational power, requiring around 5GB of memory in the worst-case scenarios. Additionally, we report memory usage for high dimensional Ant task in the Appendix A.4 (Table 5).

Table 2: Worst-case number of critics and memory consumption for D4RL locomotion datasets.

| | Critics (Quantity) | GPU Mem. (GB) |
|---|---|---|
| **SAC-N** | 500 | 5.1 |
| **EDAC** | 50 | 1.8 |
| **LB-SAC** | 50 | 5.4 |

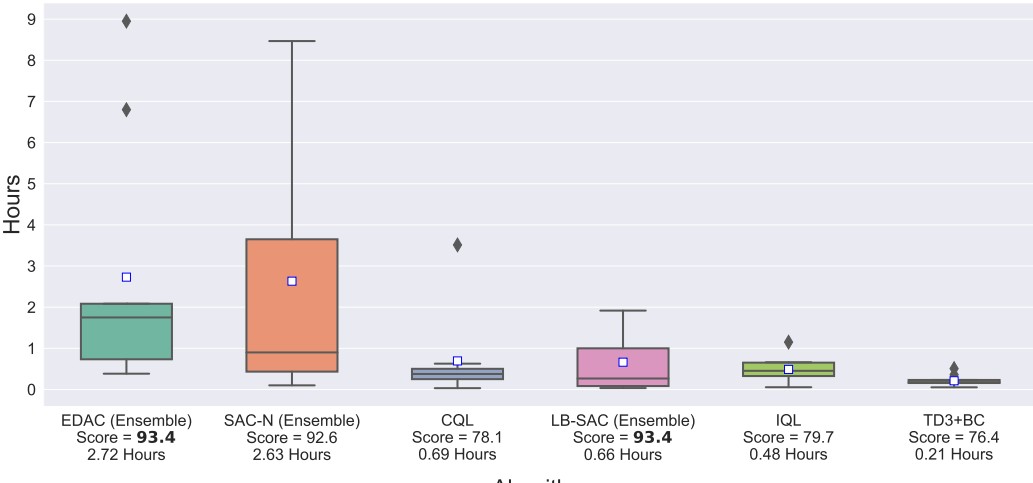

Figure 3: The convergence time of LB-SAC in comparison to other deep offline RL algorithms on 9 different D4RL locomotion datasets (Fu et al., 2020). We consider an algorithm to converge similar to (Reid et al., 2022), i.e., marking convergence at the point of achieving results similar to those reported in performance tables. White boxes denote mean values (which are also illustrated on the x-axis). Black diamonds denote samples. LB-SAC shows significant improvement distribution-wise in comparison to both SAC-N and EDAC, notably performing better on the worst-case dataset even when compared to the ensemble-free CQL algorithm. Raw values can be found in the Appendix A.6.

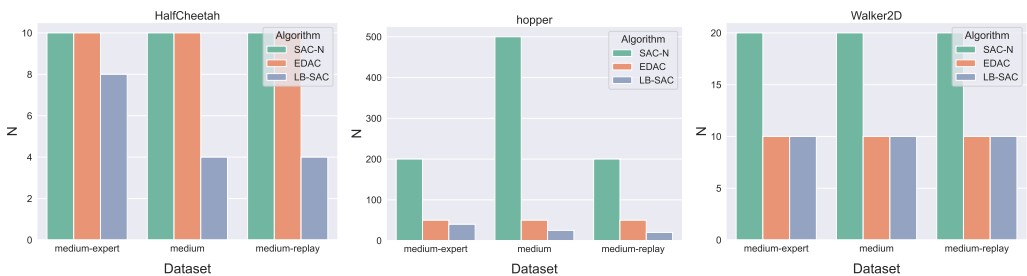

Figure 4: The number of Q-ensembles (N) used to achieve the performance reported in Table 1 and the convergence times in Figure 3. LB-SAC allows to use a smaller number of members without the need to introduce an additional optimization term. Note that, for both SAC-N and EDAC, we used the number of critics described in the appendix of the original paper (An et al., 2021).

## 5.2 Bigger Batch Sizes Penalize OOD Actions Faster

As argued in An et al. (2021), the major factor contributing to the success of Q-ensemble based methods in deep ORL setups is the penalization of actions based on prediction confidences. This is achieved by optimizing the lower bound of the value function. Both increasing the number of critics and diversifying their outputs can improve prediction confidence, which in turn leads to enhanced performance (An et al., 2021). On the other hand, unlike SAC-N, the proposed LB-SAC method does not rely on a large amount of critics (see Figure 4), or an additional optimization term for diversification as required by EDAC, while matching their performance with an improved convergence rate.

To explain the empirical success of LB-SAC, we vary the batch size and analyze the learning dynamics in a similar fashion to An et al. (2021) both in terms of out-of-distribution penalization and the resulting conservativeness, which is currently known to be the main observable property of successful ORL algorithms (Kumar et al., 2020; Rezaeifar et al., 2022). To do so, we keep track of two values. First, we calculate the standard deviation of the q-values for both random and behavioral

policies and record its relation throughout the learning process. The former is often utilized as the one producing out-of-distribution actions (Kumar et al., 2020; An et al., 2021). Second, we estimate the distance between the actions chosen by the learned policy to the in-dataset ones for tracking how conservative the resulting policies are.

The learning curves are depicted in Figure 5. We observe that increasing the batch size results in faster growth of the standard deviation relation between out-of-distribution and in-dataset actions, which can produce stronger penalization on out-of-distribution actions earlier in the training process. This, in turn, leads to an elevated level of conservativeness, as demonstrated in Figure 5b.

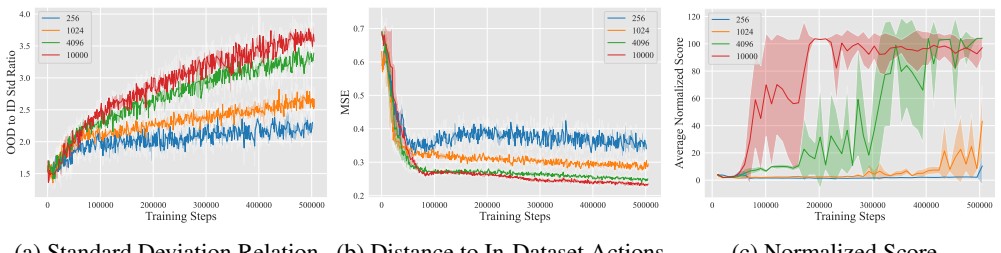

(a) Standard Deviation Relation   (b) Distance to In-Dataset Actions   (c) Normalized Score

Figure 5: Effects of increasing the batch size. **(a)** Depicts the relation of standard deviation for random actions to dataset actions **(b)** Plots the distance to the dataset actions from the learned policy **(c)** Shows averaged normalized score.

Despite the fact that Figure 5 depicts the effect of increasing batch size on standard deviation ratio between random and dataset actions, the true nature of this outcome remains not completely clear. One possible explanation can be based on a simple empirical observation confirmed by our practice (Section 2) and An et al. (2021) analysis: with further training Q-ensemble becomes more and more conservative (Figure 5a). While standard deviation for dataset actions stabilizes at some value, for random actions it continues to grow for a very long time. Hence, one might hypothesize that for different ensemble sizes $N > N'$, both will achieve some prespecified level of conservatism, but for a smaller one it will take a lot more training time. Therefore, since by increasing the batch we also increase the convergence rate, we should observe that with equal ensemble size LB-SAC will reach same penalization values but sooner.

To test the proposed hypothesis, we conduct an experiment on two datasets, fixing number of critics, only scaling batch size and learning rate as described in Section 4 and training SAC-N for 10 million instead of 1 million training steps. One can see (Figure 6) that larger batches indeed only increase convergence speed, as SAC-N can achieve same standard deviation values but much further during training. Thus, on most tasks, we can reduce the ensemble size for LB-SAC, since accelerated convergence compensates for the reduction in diversity, allowing us to remain at the same or higher level of penalization growth as SAC-N with larger ensemble.

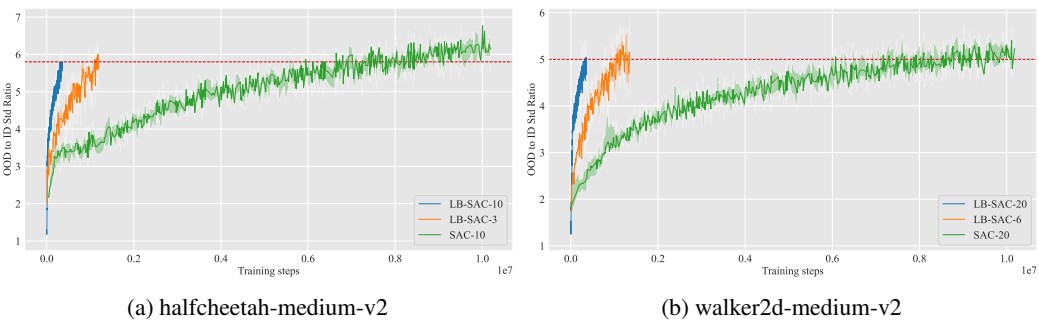

(a) halfcheetah-medium-v2                    (b) walker2d-medium-v2

Figure 6: Both LB-SAC and SAC-N with same ensemble size achieve similar standard deviation ratio between random and dataset actions, but with larger batches it happens a lot faster. This allows us to reduce the size of the ensemble for most environments while leaving the growth rate comparable to the SAC-N. Results averaged over 4 seeds. All hyperparameters except batch size and learning rate are fixed for a fair comparison. Notice that we train SAC-N for 10 million steps.

# 6 Ablations

## 6.1 How Large Should the Batch Be?

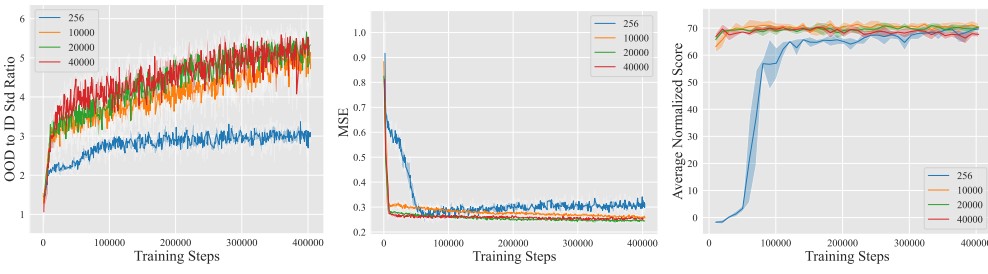

(a) Standard Deviation Relation    (b) Distance to In-Dataset Actions    (c) Normalized Score

Figure 7: Scaling batch size even further leads to diminishing returns. **(a)** Depicts the relation of standard deviation for random actions to dataset actions **(b)** Plots the distance between dataset actions and the learned policy **(c)** Shows the averaged normalized score. The number of critics is fixed for all batch sizes. Results were averaged over 4 seeds. For presentation brevity, the results are reported for the halfcheetah-medium-v2 dataset. However, similar behavior occurs on all other datasets as well.

Observing that the increased mini-batch size improves the convergence, a natural questions to ask is whether should we scale it even bigger. To answer this, we run LB-SAC with batch-sizes up to 40k.

We analyze the learning dynamics in a similar way to Section 5.2. The learning curves can be found in the Figure 7. Overall, we see that further increase leads to diminishing returns on both penalization and the resulting score. For example, Figure 7a shows that increasing the batch further leads to an improved penalization of the out-of-distribution actions. However, this improvement becomes less pronounced as we go from 20k to 40k. It may be interesting to note that the normalized scores for the typically utilized batch size of 256 start to rise after a certain level of penalization, and conservatism (MSE is around 0.3) is achieved. Evidently, the utilization of larger batch sizes helps achieve this level significantly earlier in the learning process.

## 6.2 Is It Just the Learning Rate?

To illustrate that the attained convergence rates benefit from an increased batch size and not merely from an elevated learning rate, we conduct an ablation, where we leave size of the batch equivalent to the one used in the SAC-N algorithm ($B = 256$), but scale the learning rate. The results are depicted in Figure 8.

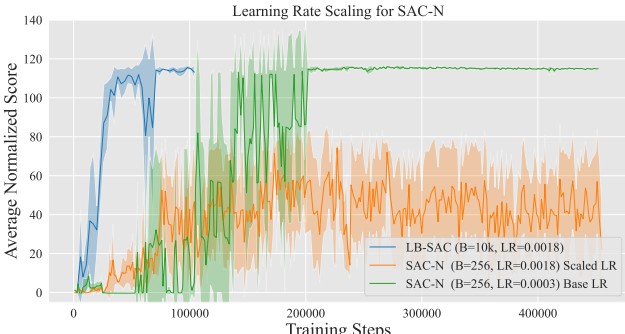

Figure 8: The average normalized score. The improvement comes from both large batch size and adjusted learning rate. Fixing batch size to the values typically used for SAC-N and scaling the learning rate does not help. The results are averaged over 4 seeds. For presentation brevity, the results are reported for the walker2d-medium-expert-v2 dataset.

Unsurprisingly, scaling learning rate without doing the same for batch size does not result in effective policies and stagnates at some point during the training process. This can be considered expected, as the hyperparameters for the base SAC algorithm were extensively tuned by an extensive number of works on off-policy RL and seem to adequately transfer to the offline setting, as described in An et al. (2021).

### 6.3 Do Layer-Adaptive Optimizers Help?

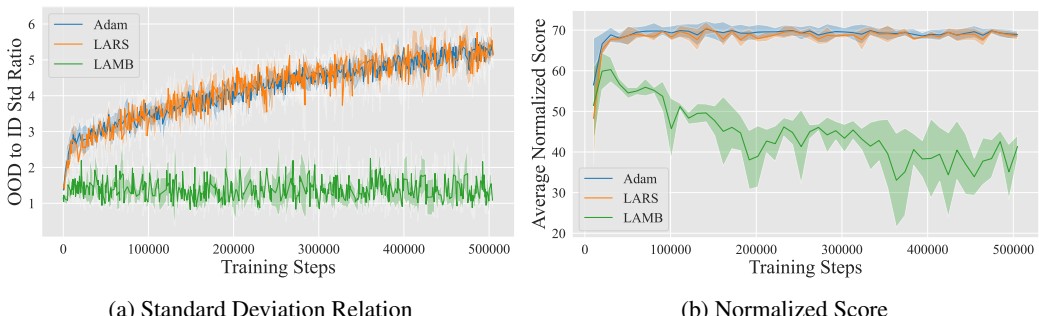

(a) Standard Deviation Relation                     (b) Normalized Score

Figure 9: Using layer-adaptive optimizers leads either to a similar performance or degrades the learning process. **(a)** Depicts the relation of standard deviation for random actions to dataset actions **(b)** Shows the averaged normalized score. The number of critics is fixed for all optimizers. The results are averaged over 4 seeds. For presentation brevity, the results are reported for the halfcheetah-medium-v2 dataset.

While we settled on a naive approach for large-batch optimization in the setting of deep offline RL, one may wonder whether using more established and sophisticated optimizers may work as well. Here, we take a look at commonly employed LARS (You et al., 2017) and LAMB (You et al., 2019) optimizers reporting learning dynamics similar to the previous sections. The resulting curves can be found in Figure 9.

We see that the LAMB optimizer could not succeed in the training process diverging with more learning steps. On the other hand, LARS behaves very similar to the proposed adjustments. While by no means we suggest that adaptive learning rates are not useful for offline RL, the straight adoption of established large-batch optimizers for typically used neural network architecture (linear layers with activations) in the context of locomotion tasks does not bring much benefits. Seemingly, these methods may require more hyperparameter tuning, different learning rate scaling rules or a warm-up schedule as opposed to the simple "square root scaling".

## 7   Conclusion

In this work, we demonstrated how the overlooked use of large-batch optimization can be leveraged in the setting of deep offline RL. We showed that a naive adjustment of the learning rate in the large-batch optimization setting is sufficient to significantly reduce the training times of methods based on Q-ensembles (4x on average) without the need to use adaptive learning rates.

Moreover, we empirically illustrated that the use of large batch sizes leads to an increased penalization of out-of-distribution actions, making it an efficient replacement for an increased number of q-value estimates in an ensemble or an additional optimization term. We hope that this work may serve as a starting point for further investigation of large-batch optimization in the setting of deep offline RL.

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

# A  Appendix

## A.1  Related Work

**Model-Free Offline RL** A large portion of recently proposed deep offline RL algorithms focuses on addressing the extrapolation issue, trying to impose a certain degree of conservatism limiting the deviation of the final policy from the behavioral one. Researchers approached this problem from multiple angles. For example, Kumar et al. (2020) proposed to directly penalize out-of-distribution actions, while Kostrikov et al. (2021) avoids estimating values for out-of-sample actions completely. Others (Fujimoto & Gu, 2021; Nair et al., 2020; Wang et al., 2020) put explicit constraints to stay closer to the behavioral policy either by directly making models closer to the behavioral one (Fujimoto & Gu, 2021), or by re-weighting behavioral policy actions with the estimated advantages. In addition, there are works that construct a latent policy action space and optimize models within it (Zhou et al., 2021; Allshire et al., 2021). In our work, we explore a different class of methods based on a model-free uncertainty quantification (An et al., 2021), and demonstrate that it can be achieved with a sufficiently large mini-batch sizes and a handful number of value estimates.

**Q-Ensembles in RL** There has been a body of work on leveraging ensembles in deep reinforcement learning (Sheikh et al., 2022; An et al., 2021; Lee et al., 2021, 2022; Osband et al., 2018). For offline RL, An et al. (2021) demonstrated that using the soft-actor critic with a large amount of value estimates leads to state-of-the art results on D4RL benchmark. As the number of critics grows, so do the computational time, in order to remedy this issue, An et al. (2021) further proposed to diversify the ensemble by disaligning the critics. In our work, we largely build upon these findings. However, we demonstrate that it is possible to alleviate a large number of critics by simply using large mini-batch sizes, which stabilizes the training process and speeds up convergence both in terms of training cycles and wall time.

**Large Batch Optimization** Unlike deep offline RL, the effect of larger mini-batch sizes was extensively studied in other areas of deep learning. This is typically referred as large batch optimization (Hoffer et al., 2017). For example, You et al. (2017) proposed to use layer-wise adaptive learning rates to train big vision models in minutes. In the Natural Language Processing field, You et al. (2019) proposed another adaptive learning rates mechanism to train attention-based models such as BERT, making it possible to train big language models within hours instead of days. In our work, we investigate a similar line of reasoning, and demonstrate that simply increasing the mini-batch size and naively adjusting the learning rate is sufficient for obtaining a faster convergence speed while matching state-of-the-art performance on the D4RL benchmark.

**Large Batch in RL** While large batches are not commonly employed in online RL, their benefits were already extensively demonstrated and analyzed. For example, Berner et al. (2019) describes a large-scale setting and uses over a million timesteps per update to train a Dota-agent. On the other hand, in simple Atari environments, batch sizes of thousands were also found to be effective for online RL algorithms Levine et al. (2017); Horgan et al. (2018); Adamski et al. (2018); Stooke & Abbeel (2018). Notably, McCandlish et al. (2018) proposed a statistic called "gradient noise scaling" to predict the largest useful batch size for both supervised and online reinforcement learning problems. As for offline RL, to the best of our knowledge, we are the first to explore the effects of large mini-batch sizes and demonstrate how it can benefit the learning process of Q-ensemble methods.

## A.2  Limitations and Future Work

**Multiple Evaluations** In our study we extensively use the notion of convergence time by evaluating multiple checkpoints during the training process. This contradicts the penultimate limitation of pure offline RL, where one should be allowed to evaluate exactly one policy. However, in this work, we were specifically interested in studying the convergence properties, therefore requiring multiple evaluations for such an analysis. Still, the practice of multiple evaluations is common in the community (Kurenkov & Kolesnikov, 2022; Seno & Imai, 2021) since the proper approach to compare offline RL methods is still an open question.

**Overfitting** It was noted in Seno & Imai (2021) that the best checkpoint across the training process significantly outperforms the final performance typically reported in deep offline RL papers. This is either attributed to the noisy learning process or some form of overfitting, as it can be often observed

that the performance starts to deteriorate with more learning steps (Seno & Imai, 2021). We observed a similar behavior for large-batch optimization on some datasets (see Section 6.1, Figure 7 for 40k batch size). We believe this is an interesting direction for future work (including large-batch settings), since the methodological apparatus for studying the properties of overfitting is yet to be developed for the offline RL setting.

**LBO for Other Offline RL Algorithms** Although the sole focus of our work was on studying large-batch optimization for Q-ensemble methods to demonstrate the benefits it can bring (e.g. a reduced number of critics or improved convergence time), an intriguing line of investigation would be the application of big batch sizes to other deep offline RL algorithms which could potentially also lead to improved convergence.

**Layeradaptive Optimizers for Offline RL** While we conducted a preliminary set of experiments on layer-adaptive optimizers (You et al., 2017, 2019) in the Appendix 6.3, more investigation is certainly needed in this direction. Since we run our experiments on D4RL benchmark and locomotion environments, we believe these optimizers may find their use in the context of different tasks typically employing other types of architecture (e.g. convolutional neural networks, recurrent networks, and transformers).

### A.3 On Layer Normalization

While scaling batch size significantly improves convergence speed and penalization of OOD actions, there were some difficult tasks where such naive approach could not help as much. As we showed in 6.1, scaling has its limits and can lead to diminishing returns, where at some point (around 10k batch size) the std ratio (as well as penalization) will stop growing. From this point, increasing batches further does not always provide much benefit, since without a sufficient level of penalization for a given task even with a faster convergence rate the agent may not converge or the training can be unstable. With a default MLP critic architecture, the only option left is to make the ensemble bigger or use diversity loss from EDAC. In our preliminary experiments, we found that increasing the ensemble size could remedy this issue but led to longer training times.

After more experimentation, we found that the use of Layer Normalization (Ba et al., 2016) completely reduces diminishing returns on all batch sizes we considered (Figure 10). One can see in Figure 10 that the scaling tendency is preserved in opposition to the results we observed without layer normalization (Figure 7).

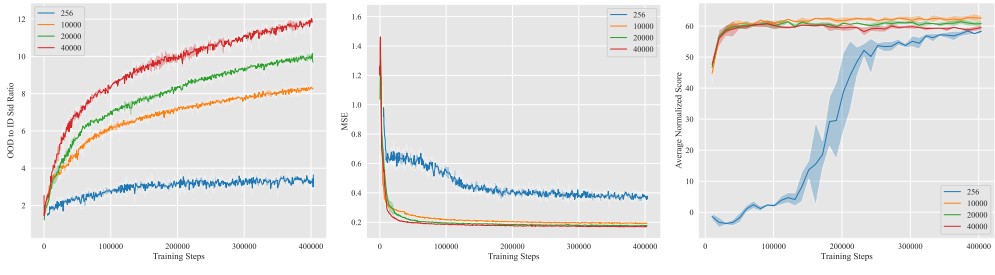

(a) Standard Deviation Relation   (b) Distance to In-Dataset Actions   (c) Normalized Score

Figure 10: Layer normalization reduces the effects of diminishing returns. **(a)** Depicts the relation of standard deviation for random actions to dataset actions **(b)** Plots the distance to the dataset actions from the learned policy **(c)** Shows the averaged normalized score. The number of critics is fixed for all batch sizes. Results were averaged over 4 seeds. halfcheetah-medium-v2 is the dataset used.

The most illustrative example is the **hopper-medium-v2** dataset, which can be considered the hardest task among all others. To solve it, SAC-N uses 500 critics while EDAC uses 50 (An et al., 2021). In our experiments, we were unable to get competitive results with bigger batches and 50 critics, and increasing ensemble size was not an option due to efficiency reasons. However, with additional layer normalization for critics (after each layer and before nonlinearity), we were able to get strong results using only 25 critics due to increased penalization. Note that on more simple tasks (Figure 10), such a boost can lead to overconservative policies and lower scores as a result (almost 10 points worse on halfcheetah-medium-v2).

## A.4 Results On Additional Datasets

Table 3: Final normalized scores on all D4RL Gym tasks, averaged over 4 random seeds. LB-SAC attains a better average performance over both the SAC-N and EDAC algorithm while being considerably faster to converge as depicted in Figure 11.

| Task Name | SAC-N | EDAC | LB-SAC |
|---|---|---|---|
| halfcheetah-random | 28.0±0.9 | 28.4±1.0 | 31.1±1.8 |
| halfcheetah-medium | 67.5±1.2 | 65.9±0.6 | 71.5±1.2 |
| halfcheetah-expert | 105.2±2.6 | 106.8±3.4 | 109.0±2.8 |
| halfcheetah-medium-expert | 107.1±2.0 | 106.3±0.6 | 109.1±2.6 |
| halfcheetah-medium-replay | 63.9±0.8 | 61.3±1.9 | 64.3±0.7 |
| halfcheetah-full-replay | 84.5±1.2 | 84.6±0.9 | 86.6±0.5 |
| hopper-random | 31.3±0.0 | 25.3±10.4 | 31.4±0.0 |
| hopper-medium | 100.3±0.3 | 101.6±0.6 | 100.7±5.2 |
| hopper-expert | 110.3±0.3 | 110.1±0.1 | 110.0±0.1 |
| hopper-medium-expert | 110.1±0.3 | 110.7±0.1 | 110.9±0.4 |
| hopper-medium-replay | 101.8±0.5 | 101.0±0.5 | 101.6±1.0 |
| hopper-full-replay | 102.9±0.3 | 105.4±0.7 | 108.3±0.3 |
| walker2d-random | 21.7±0.0 | 16.6±7.0 | 21.6±0.1 |
| walker2d-medium | 87.9±0.2 | 92.5±0.8 | 90.1±1.4 |
| walker2d-expert | 107.4±2.4 | 115.1±1.9 | 107.6±0.4 |
| walker2d-medium-expert | 116.7±0.4 | 114.7±0.9 | 113.4±3.4 |
| walker2d-medium-replay | 78.7±0.7 | 87.1±2.3 | 79.9±0.4 |
| walker2d-full-replay | 94.6±0.5 | 99.8±0.7 | 109.1±2.9 |
| Average | 84.5 | 85.2 | **86.4** |

Table 4: Normalized maximum scores on all D4RL Gym tasks, averaged over 4 random seeds. LB-SAC attains a better average performance over both the SAC-N and EDAC algorithms.

| Task Name | SAC-N | EDAC | LB-SAC |
|---|---|---|---|
| halfcheetah-random | 29.7±1.0 | 29.9±1.6 | 34.1±1.4 |
| halfcheetah-medium | 71.0±1.3 | 68.2±2.3 | 73.0±1.4 |
| halfcheetah-expert | 108.5±1.0 | 108.2±0.7 | 110.2±1.1 |
| halfcheetah-medium-expert | 109.1±2.1 | 107.0±1.3 | 110.6±2.7 |
| halfcheetah-medium-replay | 65.8±0.9 | 65.0±2.1 | 66.0±0.7 |
| halfcheetah-full-replay | 85.9±0.4 | 85.6±0.5 | 87.6±0.8 |
| hopper-random | 32.0±1.6 | 32.9±0.2 | 31.4±0.0 |
| hopper-medium | 101.0±0.6 | 102.7±0.2 | 103.8±0.0 |
| hopper-expert | 110.6±0.5 | 111.8±0.0 | 110.1±0.1 |
| hopper-medium-expert | 110.4±0.4 | 111.4±0.2 | 110.9±0.4 |
| hopper-medium-replay | 102.9±0.8 | 102.6±0.6 | 104.1±0.5 |
| hopper-full-replay | 107.9±0.3 | 107.9±0.4 | 109.3±0.3 |
| walker2d-random | 21.9±0.0 | 22.0±0.3 | 21.8±0.0 |
| walker2d-medium | 88.7±0.5 | 94.6±1.5 | 91.6±1.4 |
| walker2d-expert | 110.6±0.6 | 116.7±0.7 | 109.8±0.9 |
| walker2d-medium-expert | 116.0±0.8 | 115.3±0.2 | 115.5±1.1 |
| walker2d-medium-replay | 83.3±2.9 | 87.8±2.3 | 88.1±2.8 |
| walker2d-full-replay | 97.8±0.8 | 100.8±0.5 | 110.4±5.0 |
| Average | 86.2 | 87.2 | **88.3** |

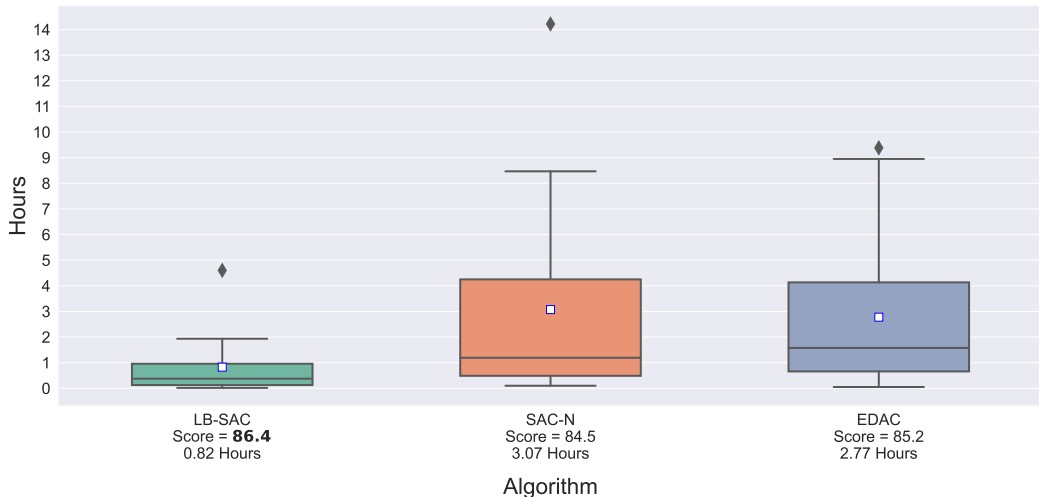

Figure 11: The convergence time of LB-SAC in comparison to SAC-N & EDAC on all D4RL Gym datasets (Fu et al., 2020). We consider an algorithm to converge similar to (Reid et al., 2022), i.e., marking convergence at the point of achieving results similar to those reported in performance tables. White boxes denote mean values (which are also illustrated on the x-axis). Black diamonds denote samples. LB-SAC shows significant improvement distribution-wise in comparison to both SAC-N and EDAC, notably performing better on the worst-case dataset.

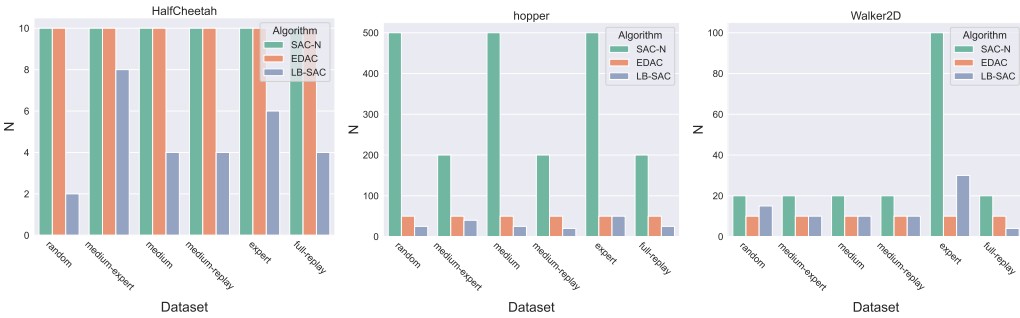

Figure 12: The number of Q-ensembles (N) used to achieve the performance reported in Table 3 and the convergence times in Figure 11. LB-SAC makes it possible to use a smaller number of members without the need to introduce an additional optimization term. Note that, for both SAC-N and EDAC, we used the number of critics described in the appendix of the original paper (An et al., 2021).

Table 5: Q-ensemble methods evaluation on ant-medium-v2 with large action and observation spaces (119 dimensions in total). Results are averaged over 4 seeds. As one can see, LB-SAC can successfully scale to high dimensional problem achieving higher score faster. Note that the EDAC's memory consumption is more than the one required by the SAC-N with higher number of critics. This is due to the need to compute the gradient for each ensemble member with respect to the action space.

| Method | Score | Critics | Time (Min) | GPU Mem. (GB) |
|--------|-------|---------|------------|---------------|
| SAC-N | 121.5±1.1 | 50 | 116 | 2.47 |
| EDAC | 120.4±1.5 | 10 | 122 | 2.85 |
| LB-SAC | **126.1±1.0** | 10 | 21 | 3.27 |

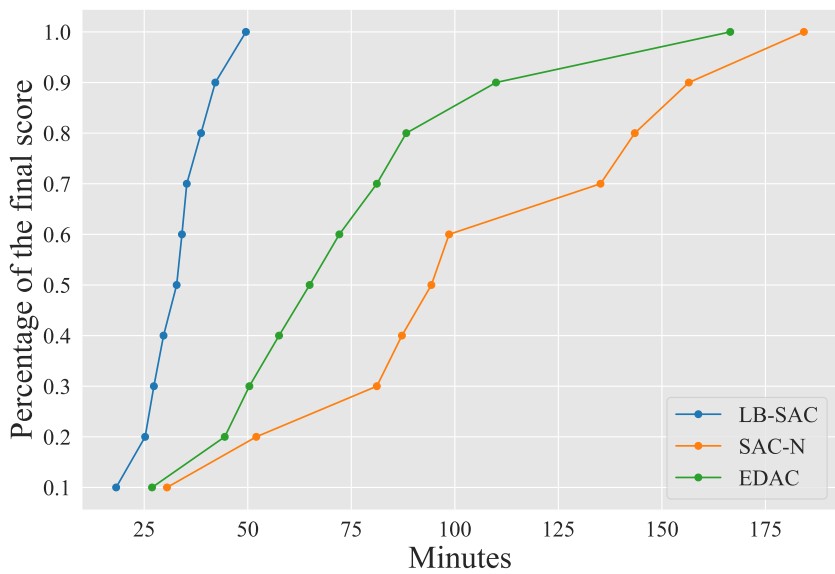

Figure 13: Relative wall-clock time in minutes to achieve a specified percentage of final score for ensemble algorithms. Results are averaged over all D4RL Gym datasets (Fu et al., 2020) with 4 seeds each. As one can see, LB-SAC converges faster to any given return threshold. Note that LB-SAC achieves competitive average final score (see Table 3).

## A.5 Hyperparameters

Table 6: SAC-N, LB-SAC, EDAC shared hyperparameters

| Parameter | Value |
|-----------|-------|
| optimizer | AdamW (Loshchilov & Hutter, 2017) |
| tau ($\tau$) | 5e-3 (5e-4 on walker2d-expert-v2) |
| hidden dim (all networks) | 256 |
| hidden layers (all networks) | 3 |
| target entropy | -action_dim |
| gamma ($\gamma$) | 0.99 |
| nonlinearity | ReLU |

Table 7: Algorithm specific hyperparameters. For **LB-SAC** algorithm on all environments we used **SAC-N** batch size and learning rate as base values for learning rate scaling (precise formula described in Equation 2).

| Algorithm | Batch size | Learning rate |
|-----------|-----------|---------------|
| SAC-N | 256 | 3e-4 |
| EDAC | 256 | 3e-4 |
| LB-SAC | 10000 | 1.8e-3 |

Table 8: Environment specific hyperparameters.

| Task Name | SAC-N (N) | LB-SAC (N, LayerNorm) | EDAC (N, $\eta$) |
|---|---|---|---|
| halfcheetah-random | 10 | 2, False | 10, 0.0 |
| halfcheetah-medium | 10 | 4, False | 10, 1.0 |
| halfcheetah-expert | 10 | 6, False | 10, 1.0 |
| halfcheetah-medium-expert | 10 | 8, False | 10, 5.0 |
| halfcheetah-medium-replay | 10 | 4, False | 10, 1.0 |
| halfcheetah-full-replay | 10 | 4, False | 10, 1.0 |
| hopper-random | 500 | 25, False | 50, 1.0 |
| hopper-medium | 500 | 25, True | 50, 1.0 |
| hopper-expert | 500 | 50, False | 50, 1.0 |
| hopper-medium-expert | 200 | 40, False | 50, 1.0 |
| hopper-medium-replay | 200 | 20, False | 50, 1.0 |
| hopper-full-replay | 200 | 25, False | 50, 1.0 |
| walker2d-random | 20 | 15, False | 10, 1.0 |
| walker2d-medium | 20 | 10, False | 10, 1.0 |
| walker2d-expert | 100 | 30, False | 10, 5.0 |
| walker2d-medium-expert | 20 | 10, False | 10, 5.0 |
| walker2d-medium-replay | 20 | 10, False | 10, 1.0 |
| walker2d-full-replay | 20 | 4, False | 10, 1.0 |
| ant-medium | 50 | 10, True | 10, 5.0 |

## A.6 Convergence Time

Table 9: Convergence times in minutes for each D4RL Gym dataset.

| Task Name | TD3+BC | IQL | CQL | SAC-N | EDAC | LB-SAC |
|---|---|---|---|---|---|---|
| halfcheetah-random | — | — | — | 18.0 | 18.0 | 1.0 |
| halfcheetah-medium | 3.1 | 6.5 | 20.6 | 26.0 | 84.0 | 5.0 |
| halfcheetah-expert | — | — | — | 435.0 | 563.0 | 50.0 |
| halfcheetah-medium-expert | 89.5 | 13.0 | 289.7 | 336.0 | 537.0 | 60.0 |
| halfcheetah-medium-replay | 89.5 | 3.2 | 24.4 | 17.0 | 23.0 | 2.0 |
| halfcheetah-full-replay | — | — | — | 83.0 | 164.0 | 4.0 |
| hopper-random | — | — | — | 7.0 | 3.0 | 19.0 |
| hopper-medium | 10.7 | 50.5 | 3.7 | 508.0 | 105.0 | 115.0 |
| hopper-expert | — | — | — | 853.0 | 276.0 | 276.0 |
| hopper-medium-expert | 10.2 | 51.0 | 58.3 | 219.0 | 408.0 | 93.0 |
| hopper-medium-replay | 11.1 | 72.2 | 112.9 | 54.0 | 44.0 | 16.0 |
| hopper-full-replay | — | — | — | 60.0 | 38.0 | 16.0 |
| walker2d-random | — | — | — | 132.0 | 73.0 | 26.0 |
| walker2d-medium | 16.1 | 11.4 | 24.4 | 39.0 | 115.0 | 15.0 |
| walker2d-expert | — | — | — | 267.0 | 330.0 | 116.0 |
| walker2d-medium-expert | 9.3 | 19.5 | 48.9 | 216.0 | 125.0 | 48.0 |
| walker2d-medium-replay | 10.2 | 10.3 | 26.3 | 6.0 | 33.0 | 4.0 |
| walker2d-full-replay | — | — | — | 41.0 | 58.0 | 26.0 |
| Average | 12.6 | 26.4 | 41.6 | 166.5 | 184.2 | 49.5 |

