# OpenReview forum: "Q-Ensemble for Offline RL: Don't Scale the Ensemble, Scale the Batch Size"
_NeurIPS.cc/2022/Workshop/Offline_RL — Offline RL Workshop NeurIPS 2022_

### Official Review · Reviewer_1v4N · 2022-10-19
**Issues with experimental methodology; Lack of clarity in important details; Unclear motivation for the presented approach;**

**Rating:** 6
**Confidence:** 4

**Review:**

Issues with experimental methodology:
* Different methods achieve different performances, so I do not understand why relative clock-time until convergence is a metric one should care about. I would imagine at least relative clock-time until a specific return threshold to make more sense?
* I have a very big issue with the author's use of "original implementations" when comparing runtimes. The same algorithm under two different implementations can have drastically different runtimes. Even something like going from Tensorflow to Jax can cause very different runtimes. Let alone all the other algorithmic components involved. For this reason, in my opinion, the runtime estimates, and in fact the performance estimates as well (due to minutiae as simple as network initialization method etc.) are completely unreliable until they are all implemented under the same implementation library.

Lack of clarity in important details:
* If you scale mini-batch from 256 --> 10K, this should be using roughly the same compute cost as using ensemble with ~40x more critics. So it is not clear to me how/why you get runtime / performance gains? In the same vein, I think in Figure 3 you should be also comparing to SAC-N/EDAC when they use ~40x more critics than LB-SAC.

Unclear motivation for the presented approach:
* In your work you present a number of results studying various effects of large batch size, and show that gains are likely due to the effect of STD amongst critics on OOD data. However, I think you should try to provide some reasoning as to why it makes sense that large batch would work better. To clarify my statement: In theory, the larger the ensemble size, the more accurate the uncertainty estimates should become (for example under the NTK interpretation of ensembles). But your results are going in the opposite direction and actually reducing the number of ensemble members. For this reason, I think your work requires more discussion -- hopefully supported with some evidence -- as to why this direction of large batch with small ensembles makes sense.

Additional Comments:
* I think you may have missed this work with seems to be directly related to the topic you are studying: https://arxiv.org/abs/2205.13703 (also they are achieving SOTA results on antmaze which generally seems to be a better testbed than gym domains since gym results are very close to saturation in the literature (e.g. refer to the table of results in the IQL paper))
* The numbers and text in in Figures 3-4 are small and hard to read.

---

### Official Review · Reviewer_HsaZ · 2022-10-20
**An interesting counterargument to prior work on Q-ensemble approaches to offline-RL**

**Rating:** 7
**Confidence:** 3

**Review:**

They present an interesting counterargument to prior work on Q-ensemble based methods for offline-RL. Their novel empirical results demonstrate that a large batch size and properly tuned learning rate can obtain competitive performance with large Q ensemble-based offline-RL methods, while remains much faster to converge. The paper presents some evaluation of this large-batch size phenomenon, showing qualitative differences in the TD error and Q values depending on the batch size. While they provide some potential explanation for their results, it is generally unclear exactly what the phenomena driving their results is and their paper would benefit from further ablation and analysis to provide a clearer explanation for their observed results.